# Beneficial effect on the soil microenvironment of *Trichoderma* applied after fumigation for cucumber production

Jiajia Wu[1], Jiahong Zhu[1], Daqi Zhang[1], Hongyan Cheng[1], Baoqiang Hao[1], Aocheng Cao[1,2], Dongdong Yan[1,2], Qiuxia Wang[1,2], Yuan Li[1,2]*

**1** Institute of Plant Protection, Chinese Academy of Agricultural Sciences, Beijing, China, **2** Beijing Innovation Consortium of Agriculture Research System, Beijing, China

* liyuancaas@126.com

**Data Availability Statement:** All relevant data are within the manuscript and its Supporting Information files.

## Abstract

Biocontrol agents applied after fumigation play an important role to the soil microenvironment. We studied the effect of *Trichoderma* applied after dimethyl disulfide (DMDS) plus chloropicrin (PIC) fumigation on the cucumber growth, soil physicochemical properties, enzyme activity, taxonomic diversity, and yield through laboratory and field experiments. The results confirmed that *Trichoderma* applied after fumigation significantly improved soil physicochemical properties, cucumber growth, soil-borne pathogens, and soil enzyme activity. Genetic analysis indicated that *Trichoderma* applied after fumigation significantly increased the relative abundance of *Pseudomonas*, *Humicola* and *Chaetomium*, and significantly decreased the relative abundance of the pathogens *Fusarium* spp. and *Gibberella* spp., which may help to control pathogens and enhanced the ecological functions of the soil. Moreover, *Trichoderma* applied after fumigation obviously improved cucumber yield (up to 35.6%), and increased relative efficacy of soil-borne pathogens (up to 99%) and root-knot nematodes (up to 96%). Especially, we found that *Trichoderma* applied after fumigation increased the relative abundance of some beneficial microorganisms (such as *Sodiomyces* and *Rhizophlyctis*) that can optimize soil microbiome. It is worth noting that with the decline in the impact of the fumigant, these beneficial microorganisms still maintain a higher abundance when the cucumber plants were uprooted. Importantly, we found one tested biocontrol agent *Trichoderma* 267 identified and stored in our laboratory not only improved cucumber growth, reduced soil-borne diseases in late cucumber growth stages but also optimized micro-ecological environment which may have good application prospect and help to keep environmental healthy and sustainable development.

## Introduction

Cucumber (*Cucumis sativus* Linn.) is one of the most extensively cultivated and consumed vegetable crops in China [1, 2]. The production of high-quality cucumber increases market sales and farmer income, and encourages further expansion of cucumber production [3].

**Funding:** Financial Disclosure Statement: This research was supported by Beijing Innovation Consortium of Agriculture Research System (BAIC01-2019), the National Key Research and Development Program of China (2018YFD0201300) and the National Natural Science Foundation of China (Program no. 31672066). Mr. Cao received these funding awards. The funders had no role in study design, data collection and analysis, decision to publish, or preparation of the manuscript.

**Competing interests:** The authors have declared that no competing interests exist.

Unfortunately, the continuous cropping on the same land depletes soil nutrients, leads to an accumulation of soil-borne pathogens and a reduction in soil microbes beneficial to crop production [4, 5]. The most effective and convenient method to control soil-borne pathogens and nematodes is soil fumigation [6]. Chloropicrin (PIC) and dimethyl disulfide (DMDS) effectively control nematodes, soil-borne pathogens such as *Fusarium* spp. and *Phytophthora* spp., and they contribute to increase crop yield [7, 8]. Although fumigants are effective and economically feasible for controlling soil-borne pathogens, their broad-spectrum impact is detrimental to all soil microbes, including those that are beneficial [9].

Soil microorganisms play an important role in the biogeochemical and nutrient cycles, organic matter formation and decomposition, and soil structure, which are known to have complex interactions amongst themselves and with crops grown in the same soil [10, 11]. The growth and quality of cucumbers may also be influenced by the relative abundance of rhizosphere microorganisms [12]. In addition, the disease presence has been closely related to changes in the soil microecology that favour plant pathogens [9]. However, after PIC fumigation, the microbial populations recover more slowly and the microbial community structure changes [13]. In addition, PIC fumigation over many years reduced the diversity of the microbial community in the soil [14] and disturbed rhizosphere microorganisms [13].

*Trichoderma* is an effective biocontrol agent for plants grown in greenhouse as well as fields, showing antifungal properties as well as promoting growth and inducing plant resistance against pathogenic microorganisms [15–18]. It is well documented that *Trichoderma* effectively controlled soil-borne pathogens such as *Fusarium* spp. and *Phytophthora* spp. [19]. Adding *Trichoderma* to fumigated soil can prolong the fumigant's control of soil-borne pathogens when populations increase over time, compared with fumigation alone which becomes ineffective against pathogens as concentrations decline over time [20, 21]. Previous research showed that inoculating *Trichoderma* sometimes failed to improve crop yield, possibly because the species inoculated without other microbial species could not adapt to the soil conditions and survive in an established microbial environment [22]. The addition of a biocontrol agent after soil fumigation may overcome some of the typical constraints of biocontrol agent or fumigation applications alone. Tian et al. and Jia et al. showed that soil fumigation followed by the application of biocontrol agents increased soil health and crop yield [23, 24]. The abundance of beneficial bacteria and the soil's microbial and physicochemical balance was also improved [25].

However, we found none that reported such taxonomic changes when two fumigants are used to fumigate the soil prior to the application of *Trichoderma*.

In this study, we first conducted indoor experiment to compare the effects of commercial *Trichoderma harzianum* and other three *Trichoderma* strains identified and stored in our laboratory, and then screened stable and efficient *Trichoderma* strains in field trials. The effects of each treatment on cucumber growth, soil-borne pathogens, soil physicochemical properties, and the changes in soil enzyme activities and microbial communities were evaluated. Moreover, we conducted dynamic monitoring of soil microbes, in order to clarify the dynamic impact on cucumber growth, soil microorganisms and ecological health and safety by applying *Trichoderma* to the soil after DMDS plus PIC fumigation.

## Materials and methods

### Soil preparation for cultivating cucumber seedlings in the laboratory

Soil was fumigated with both DMDS and PIC (DP) in a greenhouse (day/night temperature 28°C/16°C; relative humidity 50–70%; day length 10 h) in Shunyi District, Beijing (40° 13′ N, 116° 65′ E). DMDS (99.0% purity; Beijing Bailingwei Technology Co., Ltd., China) at a dosage of 60 g/m$^2$ and PIC (99.5% purity; Dalian Lvfeng Chemical Co Ltd, China) at a dosage of 20 g/

m$^2$ were artificially injected 15–20 cm into the soil and then immediately covered with 0.01 mm high-density polyethylene film (HDPE; Shandong Longxing Science and Technology Co. Ltd., China) for four weeks. Ten days after removing the film, fumigated and untreated soil were collected from within 5–20 cm of the soil surface. The soil samples were sieved through a 2 mm sieve to remove debris. The soil moisture content was adjusted to 60% of the maximum field water capacity with disinfected and deionized water. The soil samples were incubated at 28°C for 10 d in the dark. Each 300 g cultivated soil was transferred to a flowerpot for cucumber potting experiment. The physicochemical properties of the soil used in these experiments are shown in S1 Table.

## Indoor potting experiment

**Preparation of *Trichoderma* spore suspension and procedures for growing cucumber seedlings.** Commercial *Trichoderma harzianum* ('HZ'; Hainan Jinyufeng Biological Engineering Co., Ltd., China) and three other strains had been isolated, verified, and stored by our laboratory: *T. harzianum pseudoharzianum* 30 ('T30'), *T. longibrachiatum* 265 ('T265') and *T. afroharzianum* 267 ('T267'). All species and strains of *Trichoderma* were individually cultured on PDA for 5 d, then the spore suspensions were washed with sterilized and distilled water before being filtered with 4-layer gauze. The concentration of the spore suspension was adjusted to $1.0 \pm 0.05 \times 10^7$ spores/mL with a hemocytometer.

Cucumber seeds (Jingyou 4, Beijing Wanlongyufeng Seed Co., Ltd., China) were soaked in water at 60°C for 5 h, then placed on sterile wet filter paper in 150 mm diameter petri dishes at 28°C and 95% humidity, after 1 day was germination. Germinated seeds were sown into a seedling tray. At 26°C and 60% relative humidity, the cucumber seedlings were transplanted from the tray to individual pots (120 mm×130 mm) when the seedlings were in the 'three-leaf and heart' stage.

**Experimental design for seedlings cultivated in pots.** After cucumber seedlings were transplanted to the pots, the *Trichoderma* spore suspension strains were applied individually onto the cucumber seedling root. We added 30 mL *Trichoderma* spore suspension (1.0 $\pm 0.05 \times 10^7$ spores/mL) and 30 mL of commercial *Trichoderma harzianum* diluted 100 times to the soil after fumigation and without any fumigation. There were 10 treatments: DP30 (DP followed by application with *Trichoderma* strain 30), DP265, DP267, DP (fumigation only), CK30 (*Trichoderma* strain 30 without any fumigation), CK265, CK267, DPHZ (fumigation followed by commercial *T. harziamum*); CKHZ (*T. harzianum* without any fumigation); and CK (without fumigation or *Trichoderma* spp.). Each treatment contained 10 pots of cucumber seedlings. There were three applications of *Trichoderma* at intervals of 7 d. The indoor temperature and humidity were maintained at 26°C and 45%, respectively.

The cucumber seedlings were removed from the pots after six weeks. The roots were washed with water and their length were measured using calipers. The plants were air-dried in an oven with a fan at 65°C for 72 h until they no longer lost weight. A hand-held, non-destructive SPAD-502 Chlorophyll Meter (Konica Minolta, Inc., Japan) was used measure the amount of chlorophyll present in the cucumber plant leaves in each treatment. The fresh root length, stem length, stem diameter, plant fresh weight, plant dry weight and leaf chlorophyll content were recorded for plants in each treatment.

**Fungal soil-borne pathogens and root-knot nematode analysis.** Selective medium methods were used to isolate colonies of *Fusarium* spp. and *Phytophthora* spp. in the soil and to calculate their abundance, following the methods described by Komada and Masago et al., respectively [26, 27]. The size of the root-knot nematode (*Meloidogyne* spp.) population was quantified using the method described by Liu [28].

**Soil physicochemical properties and enzyme activity.** A Futura™ Continuous Flow Analytical System (Alliance Instruments, France) was used to quantify ammonia nitrogen ($NH_4^+$-N) and nitrate nitrogen ($NO_3^-N$) concentrations in each soil sample. The available phosphorus (P) was determined according to the method described by Olsen et al. [29]. Available potassium (K) was determined using a FP640 Flame Photometer (Shanghai Instruments Group Co., Ltd., China). The organic matter (OM) content was quantified according to the $K_2Cr_2O_7$-$H_2SO_4$ oxidation reduction method described by Schinner et al. [30]. A MP512-02 Precision Water Meter was used to measure the pH of the soil sample (Shanghai Sanxin Instrumentation, Inc., China). A MP513 Conductivity Meter was used to determine the electrical conductivity (EC) (Shanghai Sanxin Instrumentation, Inc., China) of the soil.

Soil sucrase and urease activities were measured as indicators of the soil's enzyme activity. Sucrase activity was measured using a Soil Saccharase (S-SC) Assay Kit (Beijing Solarbio Science & Technology Co., Ltd., China). Soil urease activity was determined by Soil Urease (S-UE) Assay Kit (Beijing Solarbio Science & Technology Co., Ltd., China). The activities of sucrase and urease were measured according to their absorbance at 630 nm and 540 nm, respectively, using a FlexStation® 3 Multimode Microplate Reader (Molecular Devices LLC., USA).

**Extraction of soil DNA, PCR amplification, high-throughput sequencing.** Total soil DNA was extracted from each 0.25 g soil sample using DNeasy Power Soil Kit (Qiagen Com., China). The extracted soil DNA was plated out onto 1% agarose gel for electrophoresis, and then the DNA concentration measured using a NanoDrop® ND-1000 UV-Vis Spectrophotometer (Thermo Fisher Scientific Inc., USA). The bacterial universal primers 338F [5'-ACTCCTACGGGAGGCAGCAG-3'] and 806R [5'-GGACTACHGGGGTWTCTAAT-3'] and fungal universal primers ITS1F [5'-CTTGGTCATTTAGAGGAAGTAA-3'] and ITS2R [5'- GCTGCTATC GATGC-3'] were used to amplify the V3-V4 region of bacteria and the ITS1 region of fungi, respectively. PCR products were detected by gel electrophoresis (plated out on 2% agarose) and purified using the EasyPure® Quick Gel Extraction Kit (TransGen Biotech Co., Ltd., China) and quantified using the QuantiFluor® dsDNA System (Fisher Scientific, USA). The purified PCR products were sequenced by Majorbio Bio-Pharm Technology Co. Ltd. (Shanghai, China) and microbial analyses were conducted using the MiSeq PE300 sequencing platform (Illumina Com., USA). The raw sequences were processed using the Mothur software. Sequences with less than 50 bp, ambiguous bases, and those with an average mass less than 20 were removed by FLASH and Trimmomatic software to obtain the effective sequences. Usearch (version 7.1 http://drive5.com/uparse/) software is used to cluster sequences with 97% similarity into Operational Taxonomic Units (OTUs). Qiime software (Version1.9.1) and Unit (v7.2) database (https://unite.ut.ee/) were used for species annotation analysis and sample community composition analysis. Qiime software (Version 1.9.1) was used to calculate the richness of the flora (Chao1 index, Shannon index) and the diversity of the flora (Simpson index, Ace index). R software (Version 2.15.3) was used to draw the dilution curve and bar diagrams of species at genus level.

## Real-time quantitative PCR

Quantitative PCR was conducted on a CFX96 Touch™ Real-Time PCR Detection System (Bio-Rad, USA) in a total volume of 20 μL. The fluorescent dye SYBR Green was used to identify the target genes. The reaction using 10 μL of 2 × SsoFast™ EvaGreen® Supermix (Bio-Rad Laboratories, USA), 1 μL of soil genomic DNA template and 0.5 μM of *Trichoderma* forward and reverse primer. Information on the *Trichoderma* gene-specific qPCR primers and thermal programs is shown in S2 Table. Melting curve analysis was used to confirm the product specificity. The amplification efficacies were > 90% and $R^2$ values were > 0.99 for the target genes.

## Field experiments

Field trials were carried out in the Shunyi District, Beijing (40˚ 13′ N, 116˚ 65′ E). The physico-chemical properties of the field soil are shown in S1 Table. The area of each plot was 1.2 m wide × 3 m long. We added 50 mL *Trichoderma* spore suspension ($1.0 \pm 0.05 \times 10^7$ spores/mL) and 50 mL of commercial *Trichoderma harzianum* diluted 100 times to the soil after fumigation and without any fumigation. There were 4 treatments: DP267 (fumigation followed by application with *Trichoderma* strain 267), DPHZ (fumigation followed by commercial *T. harziamum*), DP (fumigation only) and CK (without fumigation or *Trichoderma* spp.). Four treatments were established on randomly designed plots, with three replicates for each treatment. There were three applications of *Trichoderma* at intervals of 7 d. Soil was sampled from each treatment 2–20 cm deep on day 1 before application of *Trichoderma*, on day 7 after the third application of *Trichoderma* and when the cucumber plants were uprooted. Soil samples were refrigerated at -80˚C and 4˚C for later analysis of changes in the microbial community, soil-borne pathogens, and root-knot nematodes. The total marketable yield of cucumber from each treatment were recorded in kg at successive harvests.

## Statistical analysis

The efficacy of each treatment against soil-borne pathogens and root-knot nematode was determined by:

$$\mathrm{Y} = \frac{X_0 - X_1}{X_0} \times 100\%$$

Where Y is the relative efficacy on soil-borne pathogens and root-knot nematodes, $X_0$ is the number of soil-borne pathogens and root-knot nematodes in the control, and $X_1$ is the number of soil-borne pathogens and root-knot nematodes in the treatment group.

Data were analyzed as a one-way ANOVA using the IBM SPSS Statistics 25 software package (IBM, USA). Significant differences between treatments were identified using Duncan's new multiple range test at the 0.05 level of significance. All treatments were compared with the control (CK), except where specifically stated.

## Results

### Laboratory studies

**Changes in the plant growth.** Compared to the control, DP30, DP265 and DP267 and DPHZ significantly increased stem diameter by 44.8%, 30.6%, 54.3% and 34.1%, respectively (Table 1). In addition, DPHZ and DP30 increased chlorophyll content significantly by 25.6% and 27.6%, respectively. CK265 increased plant stem length and diameter significantly by 50.2% and 33.8%, respectively. CK30 increased stem diameter significantly by 43.0%. Compared with the single treatments, some combinations showed synergistic plant growth promotion. When *Trichoderma* were added after fumigation, especially in the DP267 treatment that increased diameter and dry weight significantly by 54.3% and 108.4%, respectively. The diameter and dry weight in DP267 increased compared to the control and DP.

DP30, DP265 or DP267 = *Trichoderma* spp. strain 30, 265 or 267 added after fumigation (see 2.2.2. in the text for detail); DPHZ = Commercial *T. harzianum* added after fumigation. DP = Fumigation without *Trichoderma*. CK30, CK265 or CK 267 = *Trichoderma* spp. strains 30, 265 or 267 added individually to soil without fumigation. CKHZ = Commercial *T. harzianum* added to soil without fumigation. CK = Untreated control. Means (N = 3) within the

**Table 1. Effect of combined fumigation with application of *Trichoderma* on the cucumber growth.**

| Treatment | Mean stem length (cm plant$^{-1}$) | | Mean stem diameter (mm plant$^{-1}$) | | Mean fresh weight (g plant$^{-1}$) | | Mean dry weight (g plant$^{-1}$) | | Mean chlorophyll content (SPAD) | |
|---|---|---|---|---|---|---|---|---|---|---|
| | Mean + SE | % | Mean + SE | % | Mean + SE | % | Mean + SE | % | Mean + SE | % |
| DP30 | 42.58±6.98a | 45.8 | 4.88±0.34a | 44.8 | 18.32±1.80b | 40.6 | 1.39±0.06abc | 67.5 | 45.22±3.41ab | 25.6 |
| DP265 | 34.02±2.06ab | - | 4.40±0.22ab | 30.6 | 18.32±1.19b | 40.6 | 1.35±0.12abc | 62.7 | 39.42±1.82bc | - |
| DP267 | 40.58±3.60ab | - | 5.20±0.19a | 54.3 | 23.77±2.25a | 82.4 | 1.73±0.18a | 108.4 | 42.04±1.68abc | - |
| DPHZ | 33.50±2.76ab | - | 4.52±0.16ab | 34.1 | 16.71±1.09bc | - | 1.31±0.12abc | 57.8 | 45.94±0.31a | 27.6 |
| DP | 36.90±3.27ab | - | 4.57±0.28ab | 35.6 | 20.24±1.85ab | 55.3 | 1.57±0.13ab | 89.2 | 39.16±1.25c | - |
| CK30 | 35.93±6.06ab | - | 4.82±0.32a | 43 | 16.40±1.52bc | - | 1.24±0.14bcd | - | 40.60±1.61abc | - |
| CK265 | 43.87±0.68a | 50.2 | 4.51±0.13ab | 33.8 | 17.80±1.78bc | - | 1.25±0.27bcd | - | 38.87±2.17c | - |
| CK267 | 37.63±1.23ab | - | 3.53±0.19c | - | 15.22±0.46bc | - | 1.03±0.08cd | - | 38.33±1.36c | - |
| CKHZ | 35.68±2.79ab | - | 3.93±0.25bc | - | 15.05±0.87bc | - | 1.06±0.09cd | - | 36.78±0.78c | - |
| CK | 29.20±2.25b | - | 3.37±0.14c | - | 13.03±0.86c | - | 0.83±0.01d | - | 36.00±1.69c | - |

same time period accompanied by the same letter were not statistically different (P = 0.05) according to Duncan's new Multiple-Range test.

**Changes in the fungal soil-borne pathogens.** Compared to the control, the number of *Fusarium* spp. and *Phytophthora* spp. were significantly reduced after DP followed by *Trichoderma* spp., achieving 96.5% to 98.9% control of both pathogens (S3 Table). Notably, DP followed by *Trichoderma* decreased *Phytophthora* spp. colonies compared to the DP, but the differences were not significant. Particularly, DP267 significantly reduced the number of *Fusarium* spp. and *Phytophthora* spp. colonies, achieving 97.7% to 98.9%.

**Changes in the soil's physicochemical properties.** The concentrations of $NH_4^+$-N and $NO_3^-$N increased significantly after DP fumigation compared to the control (S1A and S1B Fig). Available P significantly increased only in the DP, DP30, DP265, DP267 and DPHZ treatments compared with the control, but CK30, CK265, CK267 and CKHZ were similar to the control. (S1C Fig). Compared with DP, the addition of *Trichoderma* after fumigation increased the content of effective phosphorus and available potassium. Compared to the control available K was significantly increased by DP30, DP265, DP267 and DPHZ treatments (S1D Fig).

Compared with the control, the organic matter content increased significantly after the combined applications of DP and *Trichoderma*. (S1E Fig). Moreover, single applications of the *Trichoderma* did not increase the organic matter content. The soil pH decreased significantly after DP fumigation (except the DP267 treatment) compared to the control (S1F Fig). Conversely, the EC of the soil was significantly increased after applying *Trichoderma* in fumigated soil. (S1G Fig).

**Changes in the soil enzyme activity.** Compared with the control, urease activity and sucrase activity were reduced significantly by DP30, DP265, DP267, DPHZ and DP treatments (Fig 1A), but there were no significant differences in urease activity between control and the single applications of the *Trichoderma* treatments. Notably, sucrase activity was increased significantly after the combined applications of fumigants and *Trichoderma* compared with DP (Fig 1B). Particularly, DP267 significantly increased the urease activity and sucrase activity, by 45% and 114%, respectively (S4 Table).

**Changes in the abundance of *Trichoderma* in soil.** Compared to the control, the gene copy number of *Trichoderma* increased significantly when the *Trichoderma* 267 was applied, but the other three treatments were not significantly different to the control. In addition, CK267 resulted in significantly more *Trichoderma* than DP267 (S2 Fig).

**Changes in the soil's bacterial taxonomy.** After quality trimming, a total of 1,029,649 effective sequences were obtained. The average length of the effective sequences was 418 bp.

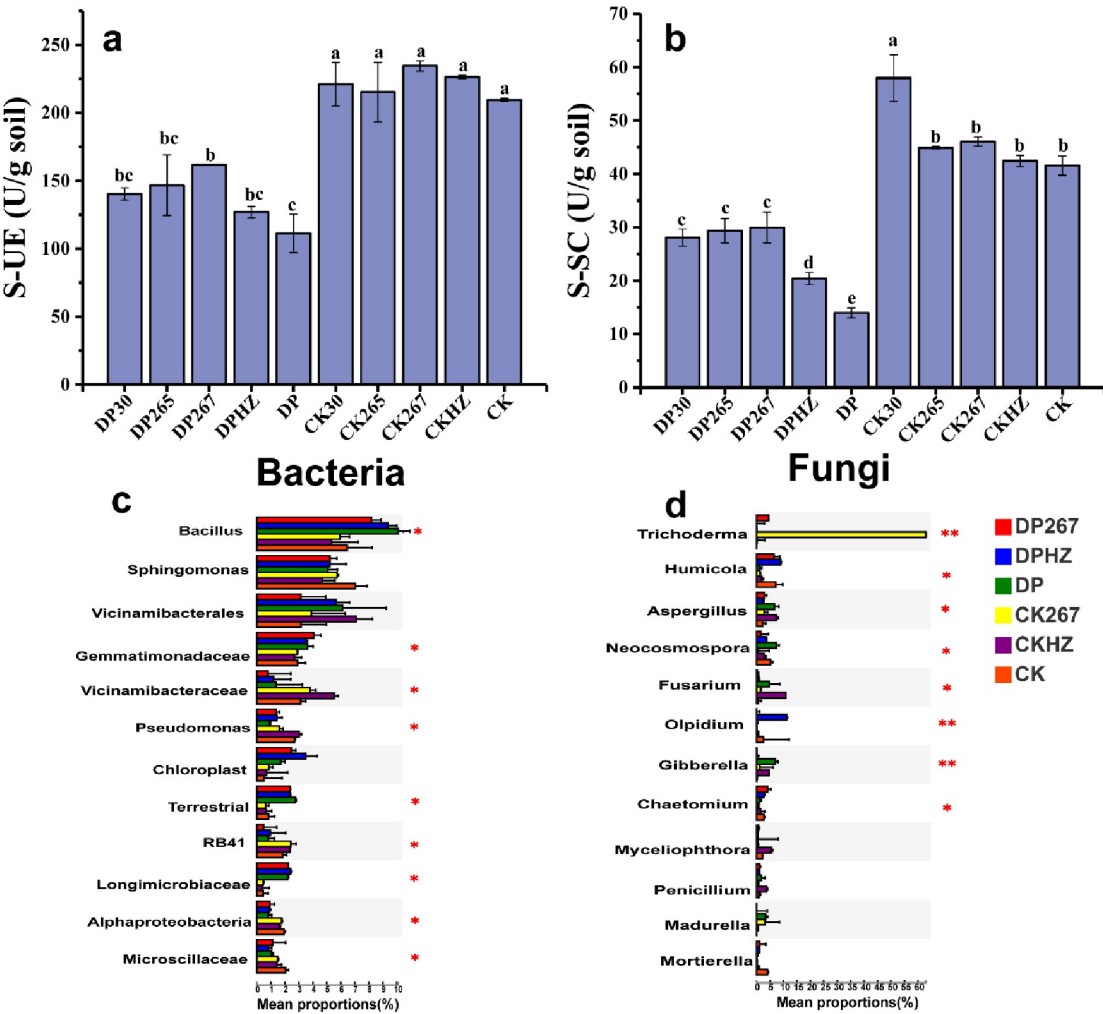

**Fig 1. Changes in soil enzyme activity and relative abundance of bacterial and fungal genera.** Soil urease (S-UE; graph a) and soil sucrase (S-SC; graph b). Means (N = 3) within the same time period accompanied by the same letter were not statistically different (P = 0.05), according to Duncan's new Multiple-Range test. Bacterial genera(c) and fungal genera (d). The number of asterisks indicates the degree of correlation (P < 0.05): (*p < 0.05, **p < 0.01, ***p < 0.001). DP30, DP265 or DP267 = *Trichoderma* spp. strain 30, 265 or 267 added after fumigation (see 2.2.2. in the text for detail); DPHZ = Commercial *T. harzianum* added to soil after fumigation. DP = Fumigation without *Trichoderma*. CK30, CK265 or CK 267 = *Trichoderma* spp. strains 30, 265 or 267 added individually to soil without fumigation. CKHZ = Commercial *T. harzianum* added to soil without fumigation. CK = Untreated control.

The rarefaction curves for bacteria reached a plateau, which indicated that the genetic data were sufficient for a reasonable. In general, the ACE and Chao1 indices indicate the community species richness, and the Shannon and Simpson indices indicate community species diversity. Biological diversity is positively correlated with Shannon, ACE and Chao1 diversity indices and negatively correlated with Simpson diversity index.

Compared to the control, the Shannon, ACE and Chao1 indices for the bacterial community in the DP, DP267 and DPHZ decreased significantly, but the Simpson diversity index increased significantly (S5 Table). Those results suggested that *Trichoderma* added to soil with DP fumigation reduced bacterial taxonomic diversity. Compared with the DP, the Shannon index of DP267 decreased significantly. There were no significant differences in the ACE and Chao1 diversity indices between the DP267 and DP.

The principal co-ordinates analysis has two main coordinate components PC1 and PC2 (S3 Fig). Species like each other are in proximity in the Principal Coordinate Analysis (PCoA) diagram. These samples from the fumigation and non-fumigation treatments could be delineated as two lineages. The non-fumigation treatments were in the direction of PC1 and separate from the fumigation treatments. PC1 and PC2 contributed 55.45% and 22.84% to the differences in species composition among treatments, respectively.

The relative abundance in the community of bacterial genera changed after fumigation. The dominant genus in the bacterial community was *Bacillus*. Compared with CK, DP267, DPHZ and DP all increased the relative abundance of *Bacillus*. The relative abundance of *Sphingomonas* decreased in each treatment compared with the control. Compared with the CK, DP267 significantly increased the relative abundance of *Gemmatimonadaceae*. DP267, DPHZ and DP all increased the relative abundance of *Pseudomonas* and *Alphaproteobacteria* compared with the control (Fig 1C).

The environmental factors were screened by the variance inflation factor (VIF). The VIF values after screening were all less than 5. Therefore, the ordination plots obtained through redundancy analysis (RDA) were used to reveal the relationship between treatments and environmental factors (S4A Fig). The results showed that pH and the relative abundance of *Trichoderma* (T) in the soil were significantly correlated with the negative of the second axis (RDA2). K, EC, and A-N were positively correlated with RDA1 and RDA2, but they were negatively correlated with pH and T.

A correlation heatmap was used to assess the relationship between bacterial abundance at the genera levels and the soil's physicochemical properties (K, electrical conductivity, nitrogen, pH) and the abundance of *Trichoderma* in soil. The results showed that bacterial abundance was affected by the soil's physicochemical properties and the abundance of *Trichoderma* in soil. There were differences in the effects of different types of bacteria on the soil's physicochemical properties and the abundance of *Trichoderma* in soil. The correlation heat map of environmental factors and 30 dominant genera of microbes showed that available K, EC, and A-N were significantly positively correlated with *Gemmatimonadaceae* and *Bacillus*; and significantly negatively correlated with *Alphaproteobacteria* (Fig 2A). In addition, A-N was significantly negatively correlated with *Pseudomonas*. pH was significantly negatively correlated with *Bacillus*, and significantly positively correlated with *Pseudomonas* and *Alphaproteobacteria*. The abundance of *Trichoderma* was significantly negatively correlated with *Bacillus*, and significantly positively correlated with *Alphaproteobacteria*.

**Changes in the soil's fungal taxonomy.** After quality trimming, a total of 1,496,147 effective reads were obtained from the genetic sequencing of fungi. The average length of the effective reads was 250 bp. The number of valid sequences detected for each soil sample exceeded 60,000 and the rarefaction curve reached a plateau, which indicated that the genetic data sufficiently represented the taxonomic composition and diversity of the fungi in the sampled soil.

Compared with the control, the Shannon index of DP267 decreased significantly, but the Simpson diversity index increased significantly (S6 Table). The ACE index of these treatments decreased significantly for all treatments except CKHZ. The Chao1 index decreased significantly for all treatments except CKHZ and DPHZ. Most of the treatments therefore decreased the diversity and richness of the soil fungal community.

The contribution of PC1 and PC2 to species composition differences between different treatment samples was 51.54% and 22.3%, respectively. The PCoA analysis delineated the treatments into three regions (S3B Fig). In relation to the abscissa, the CK267, CKHZ and DP were furthest from the control, suggesting that there was a significant difference. In relation to the ordinate, CK267 was furthest from the control, suggesting that there was a significant difference between the two samples.

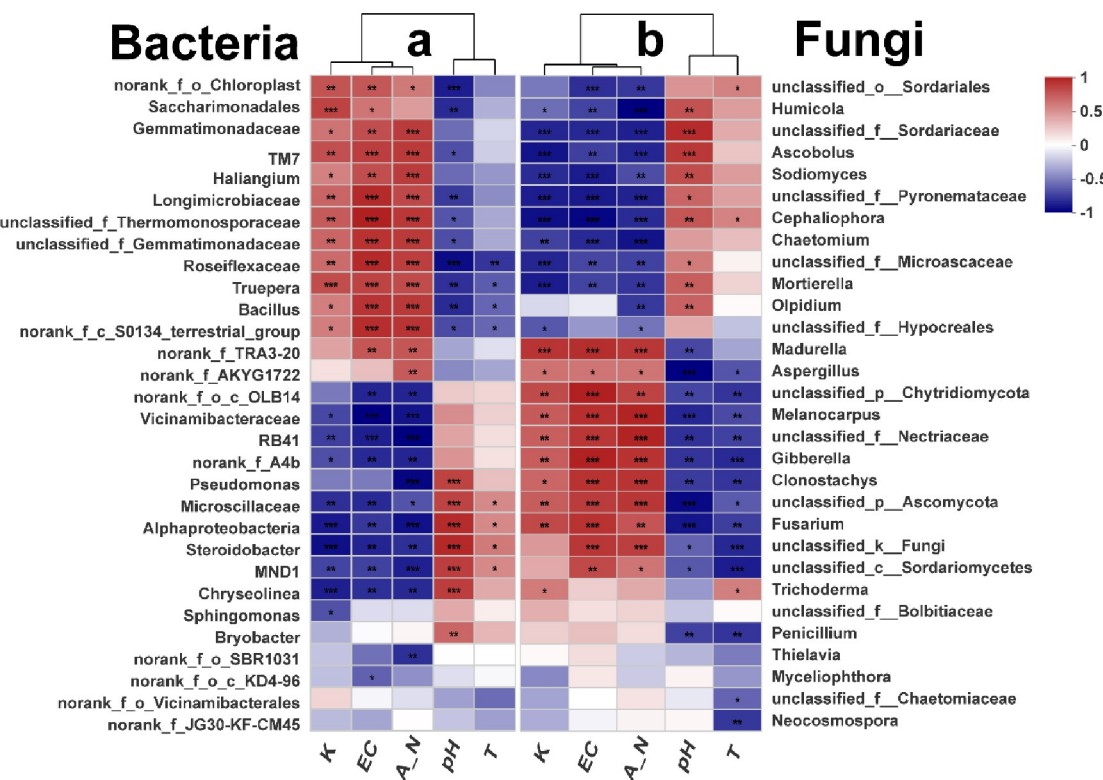

**Fig 2. Correlation heat map of environmental factors and 30 dominant genera of microbes.** pH = pH of the soil; T = The relative abundance of *Trichoderma* in the soil; K = Available potassium; A-N = Ammonium nitrogen; EC = Electrical conductivity. Different color intensities represent the normalized relative population size of each genus, based on Spearman's rank correlation coefficient. The number of asterisks indicates the degree of correlation (P < 0.05): (*p < 0.05, **p < 0.01, ***p < 0.001).

Compared with the DP, DP267 and DPHZ significantly increased the relative abundance of the genera *Humicola* and *Chaetomium*, and significantly reduced the relative abundance of *Aspergillus*, *Fusarium* and *Gibberella* (Fig 1D).

RDA results showed that pH and the presence of *Trichoderma* were significantly and positively correlated with K, EC, and A-N, but they were negatively correlated with pH and *Trichoderma* (S4B Fig).

A-N, Available K and EC was significantly negatively correlated with *Humicola* and *Chaetomium*; and significantly positively correlated with *Aspergillus*, *Gibberella*, and *Fusarium*. Soil pH was significantly positively correlated with *Humicola* and *Olpidium*; Soil pH and T significantly negatively correlated with *Aspergillus*, *Gibberella* and *Fusarium* (Fig 2B).

## Field studies

Laboratory experiment results showed that applying *Trichoderma* after fumigation can improve soil conditions, increase the relative abundance of beneficial microorganisms, and optimize the soil microenvironment. In order to verify the effectiveness of *Trichoderma* in the field, we monitored changes in soil-borne pathogens, root-knot nematodes, cucumber yield and soil's microbial community.

**Changes in the fungal soil-borne pathogens and root-knot nematodes.** When the cucumber plants were uprooted, *Trichoderma* applied after fumigation Trials 1 and 2 significantly reduced fungal soil-borne pathogens and *Meloido-gyne* spp. compared with the CK

(S7 Table). DP267 significantly reduced the number of colonies of *Fusarium* spp. by about 94.2% and 81.7% in Trials 1 and 2, respectively, whereas DP achieved only about 65.6% to 86.2% efficacy. *Trichoderma* applied after fumigation therefore improved the efficacy of *Fusarium* spp. control by 20% to 30%, compared to the DP. DP267 significantly reduced *Phytophthora* spp. by 83.5% and 95.8% in Trials 1 and 2, respectively, whereas DP achieved only about 56.1% to 80.5% efficacy. DP267 significantly reduced *Meloido-gyne* spp. by more than 96% (Trial1 and Trial 2). Compared to the DP, DP267 therefore improved the efficacy of *Meloido-gyne* spp. control by 6% to 8% (Trial 1). Compared with the CK, DPHZ had a higher percentage of control of *Fusarium* spp. than the DP267, but the percentage of control of *Phytophthora* spp. and *Meloido-gyne* spp. were lower. DP was the least effective treatment against *Fusarium* spp., *Phytophthora* spp., and *Meloido-gyne* spp.

**Changes in the cucumber yield.** Compared with the control, DP significantly increased cucumber yield by 25.1% (Trial 1), whereas DP267 and DPHZ treatments significantly increased yield by 32.7% and 35.8%, respectively. Compared with the control, DP267 and DPHZ treatments were like each other and significantly increased yield by 20% (Trial 2). Cucumber total marketable yield was 10.0% higher after application of *Trichoderma* after fumigation, compared with the DP (Fig 3 and S8 Table).

**Changes in the soil's bacterial and fungal taxonomy.** In the bacterial community, before applying *Trichoderma*, the Shannon diversity index and Chao richness index for the bacteria community fumigated with DMDS and PIC decreased significantly compared with the control (Fig 4A). After applying *Trichoderma*, the Shannon diversity index of DP267, DPHZ, DP

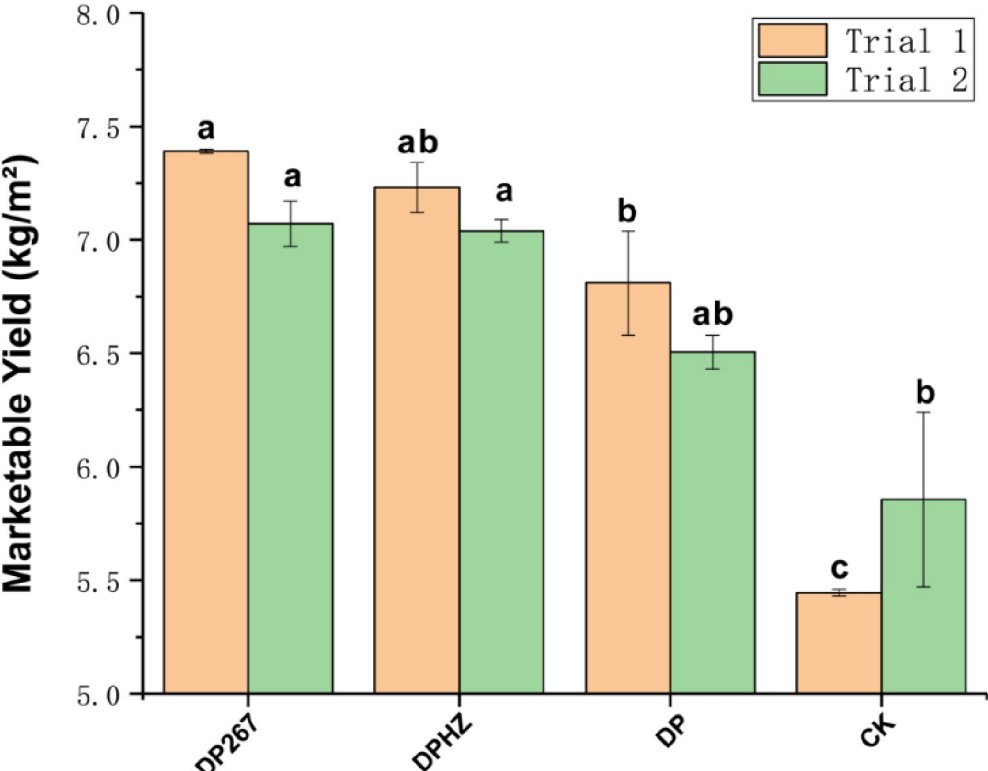

**Fig 3. The total marketable yield of cucumber.** DP267 = *Trichoderma* strain 267 added after fumigation; DPHZ = Commercial *T. harzianum* added to soil after fumigation. DP = Fumigation without *Trichoderma*. CK = Untreated control. Means (N = 3) within the same time period accompanied by the same letter were not statistically different (P = 0.05), according to Duncan's new Multiple-Range test.

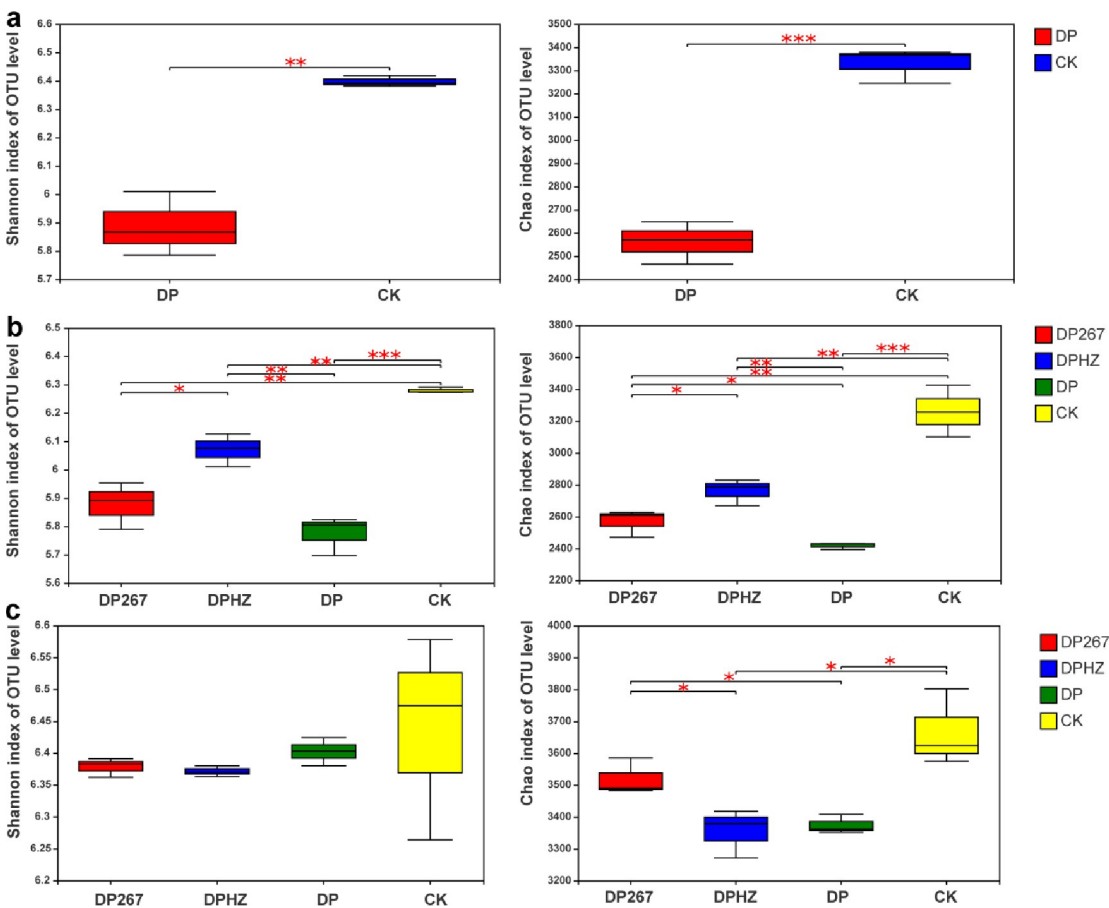

**Fig 4. Diversity analysis of bacteria in response to the following treatments.** DP267 = *Trichoderma* spp. strain 267 added to soil after fumigation. DPHZ = Commercial *T. harzianum* added to soil after fumigation. DP = Fumigation without the addition of *Trichoderma*. CK = Untreated control. Soil was sampled from each treatment 2–20 cm deep on day 1 before applying *Trichoderma* (a), on day 7 after the third applying *Trichoderma* (b) and when the cucumber plants were uprooted(c). The number of asterisks indicates the degree of correlation ($P < 0.05$): (*$p < 0.05$, **$p < 0.01$, ***$p < 0.001$).

decreased significantly compared with the control. Notably, compared with the DP, the diversity index of Shannon increased significantly in the DPHZ treatment and the richness index of Chao increased significantly in the DP267, DPHZ treatments (Fig 4B). When the cucumber plants were uprooted, there were no significant differences in the diversity index of Shannon. Notably, compared with the DP, the richness index of Chao increased significantly in the DP267, DPHZ treatments (Fig 4C).

In the fungal community, before applying *Trichoderma*, the Shannon diversity index for the fungal community fumigated with DMDS and PIC decreased significantly compared with the control, there were no significant differences in the richness index of Chao for the fungal community before applying *Trichoderma* (Fig 5A). After applying *Trichoderma*, the Shannon diversity index of DP267, DPHZ, DP decreased significantly compared with the control. Notably, compared with the DP, the diversity index of Shannon increased significantly in the DP267 treatment. The Chao richness index of DP decreased significantly (Fig 5B). When the cucumber plants were uprooted, the Shannon diversity index of DP267, DPHZ, DP increased significantly compared with the DP. Compared with the control, there were no significant differences in the richness index of Chao for the fungal community (Fig 5C). In the fungal community, DP fumigation reduced the diversity of soil microorganisms, but after adding

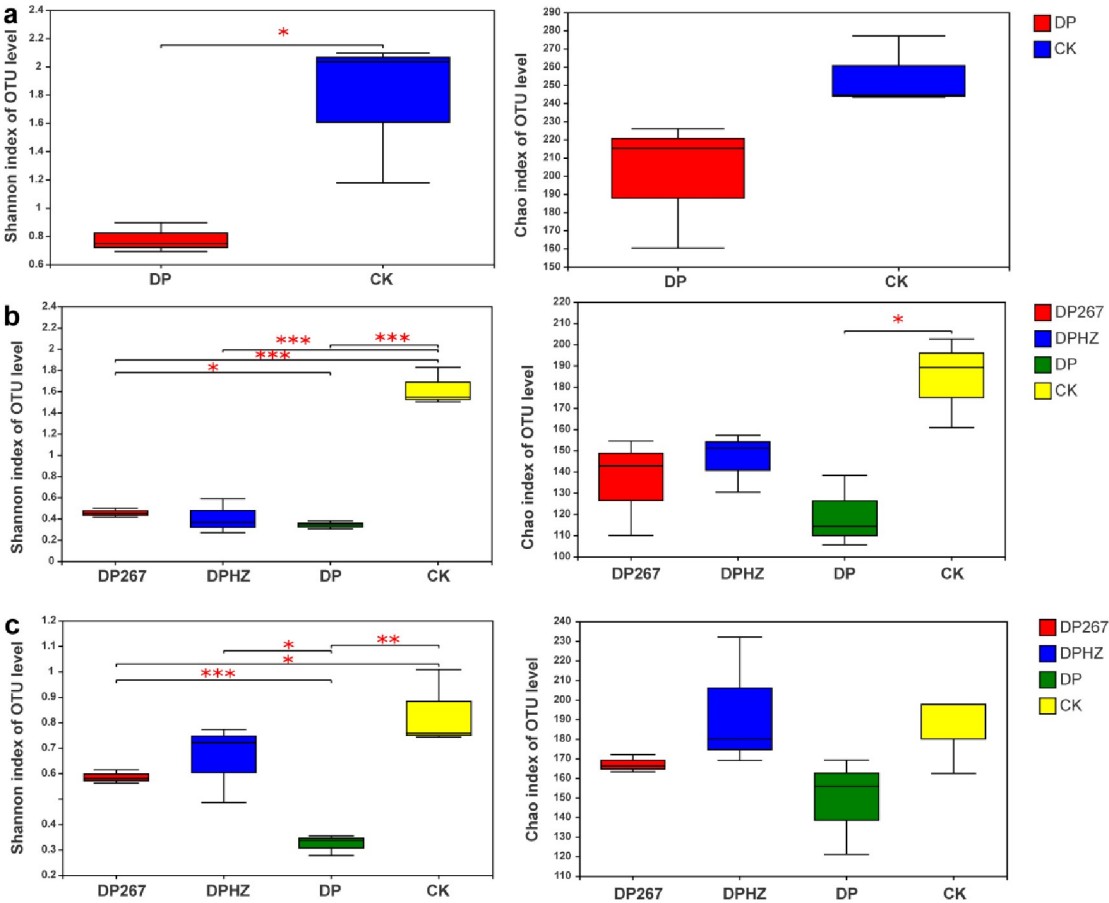

**Fig 5. Diversity analysis of fungi in response to the following treatments.** DP267 = *Trichoderma* spp. strain 267 added to soil after fumigation. DPHZ = Commercial *T. harzianum* added to soil after fumigation. DP = Fumigation without the addition of *Trichoderma*. CK = Untreated control. Soil was sampled from each treatment 2–20 cm deep on day 1 before applying *Trichoderma* (a), on day 7 after the third applying *Trichoderma* (b) and when the cucumber plants were uprooted(c). The number of asterisks indicates the degree of correlation (P < 0.05): (*p < 0.05, **p < 0.01, ***p < 0.001).

*Trichoderma*, *Trichoderma* treatments increased the diversity of microorganisms. These results indicated that *Trichoderma* can mitigate against the effects of DP fumigation by increasing the diversity of microorganisms.

There were significant differences in the relative abundance of bacterial genus before applying *Trichoderma*, after applying *Trichoderma* and when the cucumber plants were uprooted (Fig 6A–6C). During the whole growth period of cucumber, *Bacillus* is the most abundant bacterial genus. Before applying *Trichoderma*, the relative abundance of *Sphingomonas* decreased significantly compared with the control. The relative abundance of *Sphingomonas* of DP267 and DPHZ increased significantly after applying *Trichoderma* and became the second abundant bacterial genus until the cucumber plants were uprooted. Before applying *Trichoderma*, the relative abundance of *Pseudomonas* was significantly reduced, which is one of the dominant genera, and then after *Trichoderma* was added, the abundance of *Pseudomonas* decreased. The relative abundance of *Pseudomonas* eventually recovered to levels that were not significantly different.

In the fungal community, there were more genera with the significant differences that significantly changed their relative abundance after treatment with *Trichoderma* than in the

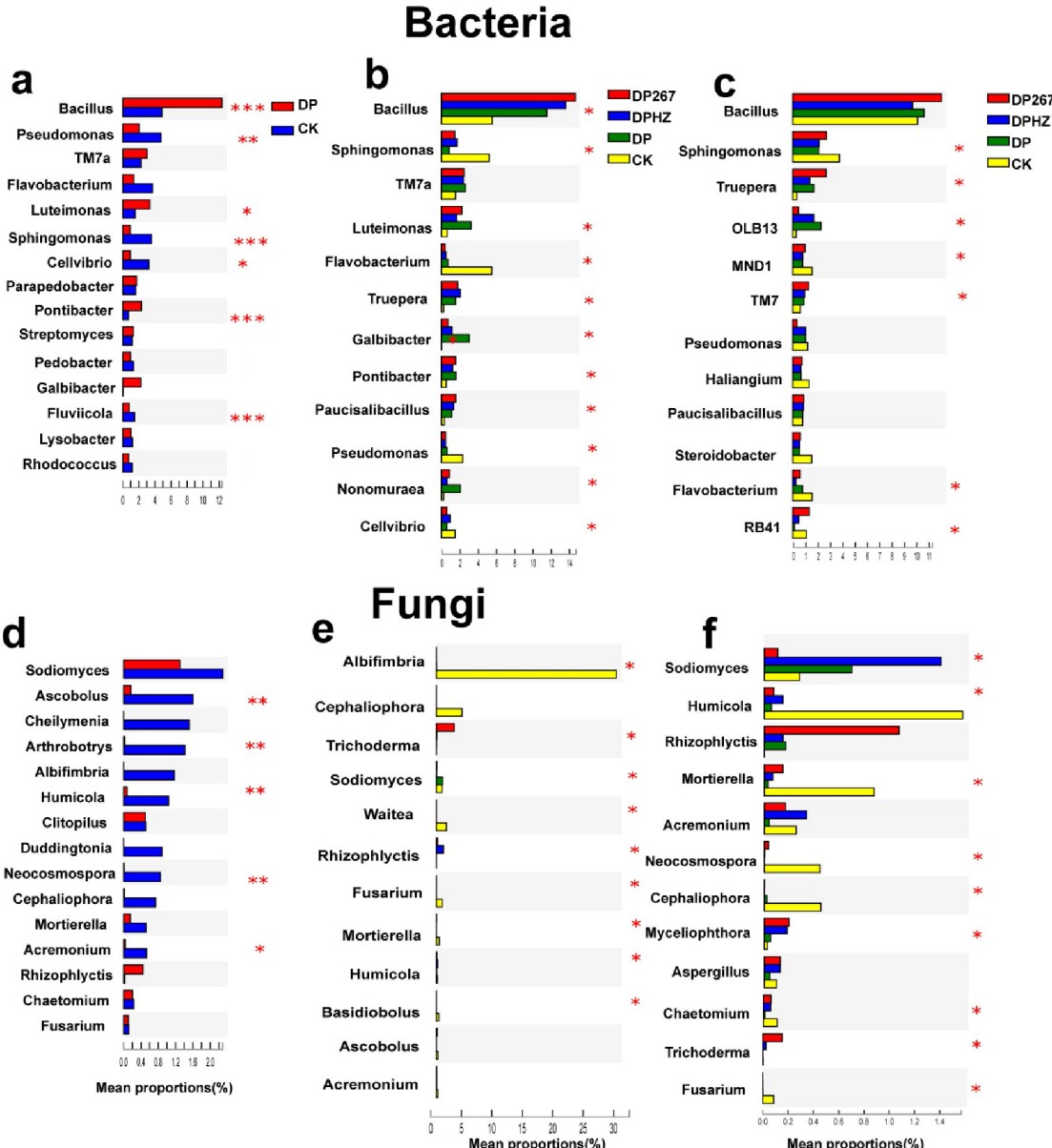

**Fig 6. Relative abundance of bacterial and fungal genera in response to the following treatments.** DP267 = *Trichoderma* spp. strain 267 added to soil after fumigation. DPHZ = Commercial *T. harzianum* added to soil after fumigation. DP = Fumigation without the addition of *Trichoderma*. CK = Untreated control. Soil was sampled from each treatment 2–20 cm deep on day 1 before applying *Trichoderma* (a, d), on day 7 after the third applying *Trichoderma* (b, e) and when the cucumber plants were uprooted (c, f). The number of asterisks indicates the degree of correlation (P < 0.05): (*p < 0.05, **p < 0.01, ***p < 0.001).

bacterial community (Fig 6D–6F). The relative abundance of some genera varied at different times. Before applying *Trichoderma*, *Sodiomyces* was the most abundant fungal genus and there was no significant difference in the relative abundance of *Sodiomyces*. After applying *Trichoderma*, *Albifimbria* became the most abundant fungal genus and the relative abundance of *Sodiomyces* decreased significantly. Compared with the control, each treatment significantly decreased the relative abundance of *Albifimbria*. When the cucumber plants were uprooted, the relative abundance of *Sodiomyces* increased significantly in the DPHZ treatment and became one of the most abundant fungal genera. Before applying *Trichoderma*, there was no

significant difference in the relative abundance of the *Rhizophlyctis* and *Chaetomium* compared with the control. After applying *Trichoderma* and when the cucumber plants were uprooted, DP267 treatment increased the relative abundance of *Rhizophlyctis*. When the cucumber plants were uprooted, *Rhizophlyctis* became one of the dominant genera, and the relative abundance of *Chaetomium* in the DP267 and DPHZ treatments significantly increased compared to DP. Compared with the control, DP267 increased the relative abundance of *Rhizophlyctis* when the cucumber plants were uprooted. After applying *Trichoderma*, each treatment significantly reduced the relative abundance of *Albifimbria*. After applying *Trichoderma* and when the cucumber plants were uprooted, fumigation treatments significantly reduced the relative abundance of *Fusarium* compared with the control. Importantly, the relative abundance of *Trichoderma* strains 267 has little change in colonization status during the growth period.

## Discussion

This study investigated the effects of *Trichoderma* applied after fumigation on the cucumber growth and soil's microecology, including changes in soil's physicochemical properties and enzyme activities, and changes in the abundance of beneficial soil microorganisms. The results suggest that *Trichoderma* applied after fumigation could improve cucumber growth and optimize soil's microecology. Importantly, more widespread use of *Trichoderma* could lead to more sustainable crop production methods by reducing the use of chemical pesticides.

### Laboratory studies

**Effects on the cucumber growth.** In this work, we found that *Trichoderma* applied after fumigation promoted cucumber growth in the laboratory. Especially, DP267 showed excellent synergistic plant growth promotion. *Trichoderma* has been reported to improve crop fitness and promote crop growth, especially when growth conditions are unfavorable [31]. *Trichoderma* can improve plant growth, which may be due to nutrient use efficiency and photosynthesis [32]. Some *Trichoderma* strains have been found to enhance plant nutrient uptake and nitrogen use efficiency [33]. Rudresh and colleagues have indicated that the application of *Trichoderma* enhance chickpea growth, nutrient uptake possibly due to an increased availability of nutrients and a release of growth promoting substances produced by the microbes [34]. Zhou et al. reported that *T. harzianum* promoted the growth of cucumber seedlings by enhancing root growth and photosynthetic capacity [35]. The application of *Trichoderma* promoted cucumber growth, enhanced the ability of cucumber to resist pathogens and may improve the disease resistance of cucumber, which has very important practical significance for increasing cucumber yield.

**Effects on the soil's physicochemical properties, soil enzyme activity and soil-borne pathogens.** Soil's physicochemical properties, enzyme activity and soil-borne pathogens in the soil are used as indicators of soil health [36]. Typical nitrification refers to the process of oxidizing $NH_3$ or $NH_4^+$ to nitrate via hydroxylamine and nitrite under aerobic conditions and the participation of nitrifying microorganisms ($NH_3/NH_4^+ \rightarrow NH_2OH \rightarrow NO_2^- \rightarrow NO_3^-$). Denitrification means that denitrifying microorganisms gradually reduce $NO_3^-$ and release $N_2$ or $N_2O$ [37]. Therefore, the nitrification of ammonia and denitrification of nitrate in soil to change the relative abundance of $NH_4^+$-N and $NO_3^-$N continuously [38]. We observed that *Trichoderma* applied after fumigation reduced nitrification and increased nitrate nitrogen in the soil, which was consistent with the results of Cheng et al. [21]. *Trichoderma* applied after fumigation increased the concentration of effective phosphorus, available potassium, and organic matter, indicating that *Trichoderma* increased soil fertility and enhanced soil

functions, which had also been reported previously [39]. Moreover, *Trichoderma* applied changed the physicochemical properties of the soil, such as increasing its ionic strength, reducing the clay, increasing the dissolved organic matter, and changing the pH [40]. We observed that some *Trichoderma* applied after fumigation treatments reduced pH but increased EC, which may be related its ability to participate in the production of metabolites such as amino acids and proteins. Soil pH had direct or indirect impacts on the abundance of microbial populations that improved the soil's antibacterial properties.

We observed that fumigation had a transient effect on soil enzyme concentration, which was consistent with the results of Zhang et al. [41]. Importantly, we observed that *Trichoderma* applied after fumigation, especially DP267 treatment, significantly increased the activity of soil enzymes, compared with the DP. *Trichoderma* added after fumigation appeared to alleviate the adverse effects of fumigation and accelerate soil recovery. Previous research reported that *Trichoderma* improved the soil environment, increased enzymes activity and root respiration [42, 43]. Soil enzymes increased in activity which improves the prospects of restoring the soil's ecological health. This will help maintain the healthy development of soil micro-ecological stability for a long time.

We observed that each treatment reduced the populations of *Fusarium* spp. And *Phytophthora* spp., indicating that *Trichoderma* or fumigation can effectively control soil pathogens. It is well documented that *Trichoderma* or fumigation effectively controlled soil-borne pathogens such as *Fusarium* spp. And *Phytophthora* spp. [19, 44].

**Effect on soil microbial community.** We found that fumigation significantly reduced the alpha diversity indices of soil bacterial community. We observed that the difference between single *Trichoderma* applied treatments and CK in the alpha diversity indices was the least significant, which indicated that single *Trichoderma* treatments had a little impact on the diversity of the bacterial communities. In the fungi community, we observed that the alpha diversity indices of abundance were largest between CK and DP267, indicating that DP267 significantly affected the diversity of fungal communities. Previous research also reported that fumigation reduced the abundance and diversity of soil bacterial and fungal communities [41]. The PcoA analysis showed that soil fumigation significantly changed the taxonomic composition of the soil's bacterial and fungal communities.

In the bacterial community, the dominant genus in the soil was *Bacillus* spp. Compared to the CK, DP267 and DPHZ increased the relative abundance of *Bacillus* and *Gemmatimonadaceae*. *Bacillus* and *Gemmatimonadaceae* are considered beneficial bacteria as plant growth improves when they are inoculated into the soil [45]. *Gemmatimonadaceae* was significantly positively correlated with available potassium, which was consistent with the study of Liang et al. [46]. The relative abundance of *Bacillus* and *Gemmatimonadaceae* was significantly positively correlated with soil nutrients. We observed that application of DP267 and DPHZ increased the abundance of *Pseudomonas* compare to the DP. *Pseudomonas* is reported to be an effective biological control agent and therefore beneficial for controlling plant pathogens [47].

In the fungi community, the relative abundance of *Humicola* and *Chaetomium* in the DP267 and DPHZ significantly increased, indicating *Trichoderma* promoted *Humicola* and *Chaetomium* population growth. *Humicola* is a common, filamentous fungi found in the soil that can decompose plant residues by secreting thermostable cellulase [48]. *Chaetomium* is a biocontrol agent that reduces pathogenic fungi and promotes plant growth [49].

The results indicated that *Trichoderma* can promote the recovery of beneficial microorganisms. These beneficial microorganisms increased the availability of essential nutrients to plants (e.g., nitrogen, phosphorus), and produce and regulate compounds involved in plant growth [50]. The presence of beneficial bacteria and fungi, which may in turn improve the

productivity and disease resistance of cucumber. In order to confirm the effect of adding *Trichoderma* after fumigation on cucumber yield and soil-borne diseases, we conducted a field experiment to evaluate the effect of *Trichoderma* in the field.

## Field studies

**Effect on soil-borne pathogens, root knot nematode and yield.**   When the cucumber plants were uprooted, our field results showed that *Trichoderma* applied after fumigation reduced soil-borne pathogens and increased the total marketable yields of cucumber, which was consistent with the results of Fang et al. [8]. DP267 significantly decreased the occurrence of root-knot nematode and increased the cucumber yield. In the later stage of cucumber growth, the efficiency of fumigation on soil-borne pathogens and root-knot nematode was reduced, but DP267 treatment still maintained a high efficiency. The results showed that the successful colonization of *Trichoderma* can continue to protect cucumbers from soil-borne diseases and reduce the use of pesticides in the later stages of cucumber growth. That may be due to the addition of *Trichoderma* can have direct and indirect effects on nematode, including inducing resistance in plant, and reducing nematode feeding and egg hatching [51]. Yan et al. found that *T. harzianum* could enhance tomato resistance to root-knot nematode by stimulating secondary metabolism and defense enzyme activity that greatly contribute to alleviate oxidative stress and suppress root-knot nematode infections [52]. Overall, the combined application reduced the abundance of soil-borne pathogens and the occurrence of root knot nematode, and improved the growing conditions for cucumbers and increased their yield. It is well documented that *Trichoderma* combined with chemical pesticides can reduce chemical pesticide application frequency, reduce soil-borne pathogens, and improve crop yield [23, 53]. Fumigation combined with *Trichoderma* treatments can strengthen the colonization of *Trichoderma* and reduce the risk of a rapid increase in the abundance of soil-borne pathogens, which is vital for reconstruction and functional restoration of soil microbial community after fumigation. It has been demonstrated that *Trichoderma* 267 exhibit promising effects against root-knot nematode.

**Effect on soil microbial community.**   Microorganisms present in the rhizosphere play a crucial role in determining the growth and health of plants and soil. The microbial interactions in the rhizosphere are often of benefit to plants, improve soil fertility, enhance the degradation of toxic chemicals. Importantly, root-associated microbiota in the rhizosphere plays important roles and positively influence the health and the growth of their host plant through various mechanisms [54]. The promotion of plant growth by microorganisms is based on a better acquisition of nutrients, hormonal stimulation and several direct or indirect mechanisms linked to plant growth, and could be involved in the reduction/suppression of plant pathogens [55, 56].

We found that *Trichoderma* to the soil after fumigation increased the diversity and richness of soil bacteria and fungi, which may have stimulated an increase in the dominant genera in the microbial community. The result was supported by previous research that reported the application of *Trichoderma* increased soil microbial diversity [57]. In addition, high soil microbial diversity or abundance can inhibit soil-borne disease pathogens [25].

After adding *Trichoderma* to DP-fumigated soil changed the bacterial and fungal community composition significantly during the growth of cucumber. After applying *Trichoderma* and when the cucumber plants were uprooted, *Bacillus* and *Sphingomonas* became dominant bacteria genus. The results indicated that after the application of *Trichoderma*, the two genera of beneficial bacteria are always in a dominant position. We opined that adding *Trichoderma* after fumigation helps the soil microbial community to rebuild. Moreover, the colonization of

*Trichoderma* can maintain the optimized soil microenvironment and protect cucumber from pathogens infection in the later stages of growth.

We found that *Trichoderma* applied after fumigation significantly changed the relative abundance of *Sodiomyces*. After applying with *Trichoderma*, the relative abundance of *Sodiomyces* decreased and then increased in the DPHZ treatment with the increased time of growth. Previous research has shown *Sodiomyces* can produce antimycotic compounds [58]. When the cucumber plants were uprooted, DP267 increased the relative abundance of *Rhizophlyctis*. *Rhizophlyctis* became one of the dominant genera. *Rhizophlyctis* is a highly effective plant biomass degrader, which can produce a diverse array of secreted enzymes [59]. These results indicated that commercial *Trichoderma* and strains had different effects on soil microorganisms, but both increased the abundance of beneficial microorganisms, and strain 267 had a better effect on cucumbers when the fumigant effect decreased.

*Trichoderma* applied after fumigation significantly changed increased the abundance of *Chaetomium* compared to DP. After applying *Trichoderma*, each treatment significantly reduced the relative abundance of *Albifimbria* and *Fusarium*. *Albifimbria* is a pathogenic fungus that can cause leaf spot on crops [60]. In addition, we found that *Trichoderma* strains 267 has little change in colonization status during the growth period, which indicated that the colonization ability of strain 267 is stronger than that of commercial *Trichoderma*. These results indicated that *Trichoderma* 267 can reduce soil-borne pathogens to increase beneficial microorganisms, rebuild soil microbial community composition, restore soil enzyme activity, and optimize cucumber rhizosphere environment.

*Trichoderma* 267 can help to decrease soil-borne pathogens, increase the abundance of beneficial microorganisms, and stabilize the soil microenvironment that is disrupted by fumigation. Many microorganisms are killed and populations reduced shortly after fumigation. *Trichoderma* can accelerate their recovery, especially when fumigant concentrations decline over time, and create a more stable soil environment in the longer term than when fumigants are used alone. We showed that *Trichoderma* also promoted the growth of cucumber, kept pathogens below economic thresholds, increased enzyme activity and the relative abundance of beneficial bacteria and fungi in the soil.

## Conclusions

Our results highlight the usefulness of *Trichoderma* strain 267 used in combination with fumigants. Compared with commercialized *Trichoderma harzianum*, *Trichoderma* 267 has the potential to become a commercial preparation. In conclusion, laboratory and field experiments have proved that *Trichoderma* applied after fumigation reduced the occurrence of soil-borne diseases, optimize the soil microenvironment, promoted cucumber growth, enhances cucumber disease resistance, and increases cucumber yield. *Trichoderma* applied after fumigation has a application prospect and help to prevent soil-borne diseases, keep environmental healthy and sustainable development.

## Supporting information

**S1 Table. Main physicochemical characteristics of laboratory and field soils.**
CK = Unfumigated soil after use in the laboratory; Field = Unfumigated field soil; DP = Soil fumigated with dimethyl disulfide and chloropicrin after it was used for growing seedlings in the laboratory.
(DOCX)

**S2 Table. The primers and thermal programs used for gene detection in this study.**
(DOCX)

**S3 Table. Effect of fumigation combined with *Trichoderma* on the percentage reduction of soil-borne pathogens.** DP30, DP265 or DP267 = *Trichoderma* strain 30, 265 or 267 added after fumigation (see 2.2.2. in the text for detail); DPHZ = Commercial *T. harzianum* added after fumigation. DP = Fumigation without *Trichoderma*. CK30, CK265 or CK 267 = *Trichoderma* strains 30, 265 or 267 added individually to soil without fumigation. CKHZ = Commercial *T. harzianum* added to soil without fumigation. CK = Untreated control. Means (N = 3) within the same time period accompanied by the same letter were not statistically different (P = 0.05) according to Duncan's new Multiple-Range test. ‡ Average cfu g$^{-1}$ soil of *Fusarium* spp. § Average cfu g−1 soil of *Phytophthora* spp.
(DOCX)

**S4 Table. Changes of soil enzyme activity following different soil treatments after fumigation.** DP30, DP265 or DP267 = *Trichoderma* strain 30, 265 or 267 added after fumigation (see 2.2.2. in the text for detail); DPHZ = Commercial *T. harzianum* added after fumigation. DP = Fumigation without *Trichoderma*. CK30, CK265 or CK 267 = *Trichoderma* strains 30, 265 or 267 added individually to soil without fumigation. CKHZ = Commercial *T. harzianum* added to soil without fumigation. CK = Untreated control. Means (N = 3) within the same time period accompanied by the same letter were not statistically different (P = 0.05) according to Duncan's new Multiple-Range test.
(DOCX)

**S5 Table. Changes in bacterial taxonomic diversity.** DP267 = *Trichoderma* strain 267 added after fumigation (see 2.2.2. in the text for detail); DPHZ = Commercial *T. harzianum* added to soil after fumigation. CK267 = *Trichoderma* strain 267 added to soil without fumigation. CKHZ = Commercial *T. harzianum* added to soil without fumigation. DP = Fumigation without *Trichoderma*. CK = Untreated control. Means (N = 3) within the same time period accompanied by the same letter were not statistically different (P = 0.05), according to Duncan's new Multiple-Range test.
(DOCX)

**S6 Table. Changes in fungal taxonomic diversity.** DP267 = *Trichoderma* strain 267 added after fumigation (see 2.2.2. in the text for detail); DPHZ = Commercial *T. harzianum* added to soil after fumigation. CK267 = *Trichoderma* strain 267 added to soil without fumigation. CKHZ = Commercial *T. harzianum* added to soil without fumigation. DP = Fumigation without *Trichoderma*. CK = Untreated control. Means (N = 3) within the same time period accompanied by the same letter were not statistically different (P = 0.05), according to Duncan's new Multiple-Range test.
(DOCX)

**S7 Table. Reduction of the number of soil-borne pathogens and root-knot nematodes in the field.** DP267 = *Trichoderma* strain 267 added after fumigation (see 2.2.2. in the text for detail); DPHZ = Commercial *T. harzianum* added to soil after fumigation. DP = Fumigation without *Trichoderma* CK = Untreated control. Means (N = 3) within the same time period accompanied by the same letter were not statistically different (P = 0.05), according to Duncan's new Multiple-Range test. ‡ Average cfu g$^{-1}$ soil of *Fusarium* spp. § Average cfu g−1 soil of *Phytophthora* spp. ‡ Average number of root-knot nematode (*Meloidogyne* spp.) per 100 g soil.
(DOCX)

**S8 Table. The total marketable yield of cucumber.** DP267 = *Trichoderma* strain 267 added after fumigation; DPHZ = Commercial *T. harzianum* added to soil after fumigation. DP = Fumigation without *Trichoderma*. CK = Untreated control. Means (N = 3) within the same time period accompanied by the same letter were not statistically different (P = 0.05), according to Duncan's new Multiple-Range test.
(DOCX)

**S1 Fig. Changes in soil physical and chemical properties.** Soil Nitrogen (A-N, graph a; and N-N, graph b), Phosphorous (P, graph c), Potassium (K, graph d), Organic matter (OM; graph e), pH (graph f)), and Soil electrical conductivity (EC; graph g). DP30, DP265 or DP267 = *Trichoderma* spp. strain 30, 265 or 267 added after fumigation (see 2.2.2. in the text for detail); DPHZ = Commercial *T. harzianum* added to soil after fumigation. DP = Fumigation without *Trichoderma*. CK30, CK265 or CK267 = *Trichoderma* spp. strains 30, 265 or 267 added individually to soil without fumigation. CKHZ = Commercial *T. harzianum* added to soil without fumigation. CK = Untreated control. Means (N = 3) within the same time period accompanied by the same letter were not statistically different (P = 0.05), according to Duncan's new Multiple-Range test.
(TIF)

**S2 Fig. Total gene copy number of *Trichoderma* in soil.** DP267 = *Trichoderma* strain 267 added after fumigation; DPHZ = Commercial *T. harzianum* added to soil after fumigation. CK267 = *Trichoderma* strain 267 added to soil without fumigation. CKHZ = Commercial *T. harzianum* added to soil without fumigation. DP = Fumigation without *Trichoderma*. CK = Untreated control. Means (N = 3) within the same time period accompanied by the same letter were not statistically different (P = 0.05), according to Duncan's new Multiple-Range test.
(TIF)

**S3 Fig. Changes in the taxonomic bacterial and fungal diversity at the OUT level.** DP267 = *Trichoderma* strain 267 added after fumigation (see 2.2.2. in the text for detail); DPHZ = Commercial *T. harzianum* added to soil after fumigation. CK267 = *Trichoderma* strain 267 added to soil without fumigation. CKHZ = Commercial *T. harzianum* added to soil without fumigation. DP = Fumigation without *Trichoderma*. CK = Untreated control.
(TIF)

**S4 Fig. Ordination plots by redundancy analysis (RDA) revealed the relationship between treatments and environmental factors.** pH = pH of the soil; T = The relative abundance of *Trichoderma* in soil; K = Available potassium; A-N = Ammonium nitrogen; EC = Electrical conductivity. The treatments were: DP267 = *Trichoderma* strain 267 added to soil after fumigation. DPHZ = Commercial *T. harzianum* strain added to soil after fumigation. DP = Fumigation without the addition of *Trichoderma*. CK267 = *Trichoderma* strain 267 added to unfumigated soil. CKHZ = Commercial *T. harzianum* strain added to unfumigated soil. CK = Untreated control.
(TIF)

## Acknowledgments

We thank Dr. Tom Batchelor for providing editorial comments and for revising the manuscript.

## Author Contributions

**Conceptualization:** Yuan Li.

**Data curation:** Jiajia Wu, Jiahong Zhu, Hongyan Cheng.

**Investigation:** Jiajia Wu, Jiahong Zhu, Daqi Zhang, Hongyan Cheng, Baoqiang Hao.

**Methodology:** Jiajia Wu, Daqi Zhang, Hongyan Cheng, Baoqiang Hao, Dongdong Yan, Qiuxia Wang.

**Project administration:** Aocheng Cao.

**Resources:** Jiajia Wu, Aocheng Cao.

**Software:** Jiajia Wu, Baoqiang Hao.

**Supervision:** Jiajia Wu, Aocheng Cao, Dongdong Yan, Qiuxia Wang, Yuan Li.

**Validation:** Dongdong Yan, Qiuxia Wang, Yuan Li.

**Visualization:** Yuan Li.

**Writing – original draft:** Jiajia Wu.

**Writing – review & editing:** Yuan Li.

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
