## [Decision Letter · Decision Letter 0]

18 Feb 2022

PONE-D-22-00468Beneficial effect on the soil microenvironment of Trichoderma applied after fumigation for cucumber productionPLOS ONE

Dear Dr. Li,

Thank you for submitting your manuscript to PLOS ONE. After careful consideration, we feel that it has merit but does not fully meet PLOS ONE’s publication criteria as it currently stands. Therefore, we invite you to submit a revised version of the manuscript that addresses the points raised during the review process. Please submit your revised manuscript by Apr 04 2022 11:59PM. If you will need more time than this to complete your revisions, please reply to this message or contact the journal office at plosone@plos.org. Please include the following items when submitting your revised manuscript:A rebuttal letter that responds to each point raised by the academic editor and reviewer(s). You should upload this letter as a separate file labeled 'Response to Reviewers'.A marked-up copy of your manuscript that highlights changes made to the original version. You should upload this as a separate file labeled 'Revised Manuscript with Track Changes'.An unmarked version of your revised paper without tracked changes. You should upload this as a separate file labeled 'Manuscript'.

We look forward to receiving your revised manuscript.

Kind regards,

Rashid Nazir

Academic Editor

PLOS ONE

Journal Requirements:

2. During your revisions, please confirm whether the wording in the title is correct and update it in the manuscript file and online submission information if needed. Specifically, the title appears as "Type of contribution: Research Paper" on the online submission information.

"This research was supported by Beijing Innovation Consortium of Agriculture Research System (BAIC01-2019), the National Key Research and Development Program of China (2018YFD0201300) and the National Natural Science Foundation of China (Program no. 31672066). We thank Dr. Tom Batchelor for providing editorial comments and for revising the manuscript."

We note that you have provided funding information. However, funding information should not appear in the Acknowledgments section or other areas of your manuscript. We will only publish funding information present in the Funding Statement section of the online submission form. 

"Financial Disclosure Statement:

This research was supported by Beijing Innovation Consortium of Agriculture Research System (BAIC01-2019), the National Key Research and Development Program of China (2018YFD0201300) and the National Natural Science Foundation of China (Program no. 31672066). 

Mr. Cao received these funding awards.

4. Please include a copy of Table 1 which you refer to in your text on page 13.

Additional Editor Comments:

Please ensure the ms format as per journal's requirements.

Reviewers' comments:

Reviewer's Responses to Questions

**Comments to the Author**

1. Is the manuscript technically sound, and do the data support the conclusions?

Reviewer #1: Yes

Reviewer #2: Yes

2. Has the statistical analysis been performed appropriately and rigorously? 

Reviewer #1: Yes

Reviewer #2: Yes

3. Have the authors made all data underlying the findings in their manuscript fully available?

Reviewer #1: Yes

Reviewer #2: Yes

4. Is the manuscript presented in an intelligible fashion and written in standard English?

Reviewer #1: Yes

Reviewer #2: Yes

5. Review Comments to the Author

Reviewer #1: This manuscript aims investigated the effects of Trichoderma applied after fumigation on the cucumber growth and soil's microecology, including changes in soil’s physicochemical properties and enzyme activities, and changes in the abundance of beneficial soil microorganisms. The manuscript was very well written, with robust results indicating that the study was well carried out. I suggest that the authors make some small changes to the manuscript, in order to improve it. Some information was difficult to read in the Figures. Standardize units, references and abbreviations.

Reviewer #2: This manuscript is generally well-written and the work was well-executed and informative. There are some points, which need to be clarified, which I have indicated directly in the edited version attached to the manuscripts.

The title, abstract and keywords are enough to present what was searched and the main results.

The introduction is well written and justifies what was done in the research.

Materials and methods are well written. However, some adjustments should be made for better understanding by the reader (suggestions in the comments in the document).

Overall, the discussion is well written and explains the results. However, some points should be detailed for better understanding by the reader (check in the comments)

Conclusions are ok.

6. PLOS authors have the option to publish the peer review history of their article (what does this mean?). If published, this will include your full peer review and any attached files.

Reviewer #1: **Yes: **Angélica Santos Rabelo de Souza Bahia

Reviewer #2: No

---

## [Author Response · Author response to Decision Letter 0]

25 Feb 2022

Dear Editor Rashid Nazir,

Thank you very much for your letter and suggestions. We have revised the manuscript. We would like to resubmit it for your consideration. We have addressed the comments raised by the reviewers. Point by point responses to the reviewers’ comments are listed below this letter. We hope that the revised manuscript is now acceptable for publication in your journal.

With best wishes,

Yours sincerely,

Yuan Li

We would like to express our sincere thanks to you and the reviewers for the careful and valuable comments on the manuscript.

Replies to the reviewers:

Response to reviewer #2: 

1. Add air temperature and relative humidity.

Response: 

Thank you for your comments. We have added air temperature and relative humidity according to your suggestion. Soil was fumigated with both DMDS and PIC (DP) in a greenhouse (day/night temperature 28℃/16℃; relative humidity 50−70%; day length 10 h) in Shunyi District, Beijing (40° 13′ N, 116° 65′ E). We have added this part in line 5-6 on page 6 of Materials and methods of the manuscript.

2. Add after how many days was germination.

Response: 

Thank you for your comments. We have added “how many days was germination” according to your suggestion. Cucumber seeds (Jingyou 4, Beijing Wanlongyufeng Seed Co., Ltd., China) were soaked in water at 60°C for 5 h, then placed on sterile wet filter paper in 150 mm diameter petri dishes at 28℃ and 95% humidity, after 1 day was germination. We have added this part in line 16 on page 7 of Materials and methods of the manuscript.

3. Detail the climatic conditions at the time of transplantation.

Response: 

Thank you for your comments. At 26℃ and 60% relative humidity, the cucumber seedlings were transplanted from the tray to individual pots (120 ×130 mm) when the seedlings were in the ‘three-leaf and heart’ stage. We have added this part in line 15 on page 7 of Materials and methods of the manuscript.

4. Add the volume of the pots.

Response: 

Thank you for your comments. At 26℃ and 60% humidity, the cucumber seedlings were transplanted from the tray to individual pots (120 ×130 mm) when the seedlings were in the ‘three-leaf and heart’ stage. We have added the volume of the pots in line 18 on page 7 of Materials and methods of the manuscript.

5. Detail how the pigments were determined.

Response: 

Thank you for your comments. A hand-held, non-destructive SPAD-502 Chlorophyll Meter (Konica Minolta, Inc., Japan) was used measure the amount of chlorophyll present in the cucumber plant leaves in each treatment. We have described this part in greater detail in line 18-21 on page 8 of Materials and methods of the manuscript.

6. Authors should explain why Trichoderma provided plant growth. In relation, the greater availability of nutrients and, consequently, greater possibility of plants to absorb forms of N and P. Chlorophyll contents should be explored.

Response: 

Thank you for your comments. Trichoderma can improve plant growth, which may be due to nutrient use efficiency and photosynthesis. Some Trichoderma strains have been found to enhance plant nutrient uptake and nitrogen use efficiency. Rudresh and colleagues have indicated that the application of Trichoderma enhance chickpea growth, nutrient uptake possibly due to an increased availability of nutrients and a release of growth promoting substances produced by the microbes. Zhou et al. reported that T. harzianum promoted the growth of cucumber seedlings by enhancing root growth and photosynthetic capacity. We have explained this part on page 28-29 of Discussion of the manuscript.

7. Authors should explain why the nitrification of ammonia and denitrification of nitrate in soil is reported to change the relative abundance of NH4+ -N and NO3-N continuously.

Response: 

Thank you for your comments. Typical nitrification refers to the process of oxidizing NH3 or NH4+ to nitrate via hydroxylamine and nitrite under aerobic conditions and the participation of nitrifying microorganisms (NH3/NH4+→NH2OH→NO2−→NO3−). Denitrification means that denitrifying microorganisms gradually reduce NO3- and release N2 or N2O. Therefore, the nitrification of ammonia and denitrification of nitrate in soil to change the relative abundance of NH4+-N and NO3--N continuously. We have explained this part in line 13-20 on page 29 of Discussion of the manuscript.

8. Authors should describe in greater detail how the use of Trichoderma reduces the presence of nematodes.

Response: 

Thank you for your comments. That may be due to the addition of Trichoderma can have direct and indirect effects on nematode, including inducing resistance in plant, and reducing nematode feeding and egg hatching. Yan et al. found that T. harzianum could enhance tomato resistance to root-knot nematode by stimulating secondary metabolism and defense enzyme activity that greatly contribute to alleviate oxidative stress and suppress root-knot nematode infections. We have described this part in greater detail in line 15-21 on page 33 of Discussion of the manuscript.

---

## [Decision Letter · Decision Letter 1]

21 Mar 2022

Beneficial effect on the soil microenvironment of Trichoderma applied after fumigation for cucumber production

PONE-D-22-00468R1

Dear Dr. Li,

We’re pleased to inform you that your manuscript has been judged scientifically suitable for publication and will be formally accepted for publication once it meets all outstanding technical requirements.

Kind regards,

Rashid Nazir

Academic Editor

PLOS ONE

Additional Editor Comments (optional):

Reviewers' comments:

Reviewer's Responses to Questions

**Comments to the Author**

1. If the authors have adequately addressed your comments raised in a previous round of review and you feel that this manuscript is now acceptable for publication, you may indicate that here to bypass the “Comments to the Author” section, enter your conflict of interest statement in the “Confidential to Editor” section, and submit your "Accept" recommendation.

Reviewer #1: All comments have been addressed

2. Is the manuscript technically sound, and do the data support the conclusions?

Reviewer #1: Yes

3. Has the statistical analysis been performed appropriately and rigorously? 

Reviewer #1: Yes

4. Have the authors made all data underlying the findings in their manuscript fully available?

Reviewer #1: Yes

5. Is the manuscript presented in an intelligible fashion and written in standard English?

Reviewer #1: Yes

6. Review Comments to the Author

Reviewer #1: (No Response)

7. PLOS authors have the option to publish the peer review history of their article (what does this mean?). If published, this will include your full peer review and any attached files.

Reviewer #1: **Yes: **Angélica Santos Rabelo de Souza Bahia

---

## [Editor Report · Acceptance letter]

25 Mar 2022

PONE-D-22-00468R1 

Beneficial effect on the soil microenvironment of *Trichoderma* applied after fumigation for cucumber production 

Dear Dr. Li:

I'm pleased to inform you that your manuscript has been deemed suitable for publication in PLOS ONE. Congratulations! Your manuscript is now with our production department. 

Kind regards, 

on behalf of

Dr Rashid Nazir 

Academic Editor

PLOS ONE